# Physiological and Population Responses of *Nilaparvata lugens* after Feeding on Drought-Stressed Rice

**DOI:** 10.3390/insects13040355

**Published:** 2022-04-05

**Authors:** Xinyan Liang, Lin Chen, Xiaoying Lan, Guangrong Liao, Ling Feng, Jitong Li, Wenyan Fan, Shuang Wang, Jinglan Liu

**Affiliations:** 1College of Horticulture and Plant Protection, Yangzhou University, Yangzhou 225009, China; mx120200774@yzu.edu.cn (X.L.); chenlin88@yzu.edu.cn (L.C.); xy2249628059@163.com (X.L.); mz120201300@yzu.edu.cn (G.L.); mz120190995@yzu.edu.cn (L.F.); dx120200130@yzu.edu.cn (J.L.); fanwy0406@163.com (W.F.); wshhaitun@126.com (S.W.); 2Joint International Research Laboratory of Agriculture and AgriProduct Safety of the Ministry of Education, Yangzhou University, Yangzhou 225009, China

**Keywords:** drought stress, rice, *Nilaparvata lugens*, physiology, biochemistry, ultrastructure of flight muscles

## Abstract

**Simple Summary:**

Drought is considered a critical threat to crop growth and sustainable agriculture worldwide, and it also greatly impacts insect development and population growth. Brown planthopper (BPH), *Nilaparvata lugens* (Stål), is the predominant rice crop pest in China, and the damaging effects of BPH are enhanced by its strong migratory and reproductive capacities. Our results provide useful information about the effect of drought stress on the poor population growth and negative physiological changes in BPH. Negative changes to water balance and osmotic pressure can cause a decline in the quality of BPH; the GST content of BPH feeding on drought-stressed rice was significantly higher than BPH feeding on non-stressed control plants, and the length of flight muscle sarcomeres and mitochondrial content were decreased in BPH feeding on drought-stressed rice. These findings suggest that water management greatly impacts the physiology and population growth of BPH, and provide a basis for understanding physiological and population-wide responses in BPH during drought stress, which may be helpful in understanding the relationship between drought stress and BPH infestation.

**Abstract:**

Drought stress greatly impacts insect development and population growth. Some studies have demonstrated increased reproductive capacity in drought-stressed insects; however, physiological changes in the brown planthopper (BPH), *Nilaparvata lugens* (Stål), during periods of drought are unclear. In this study, BPH fed on drought- stressed rice had lower population numbers than BPH feeding on non-stressed rice. Water content, osmotic pressure of hemolymph and total amino acid content of BPH were significantly lower when BPH fed on drought-stressed rice compared to the non-stressed control; however, glucose content and glutathione S-transferase (GST) activity were significantly higher in BPH fed on drought-stressed rice. The expression of *Vitellogenin* and *Exuperantia* in BPH fed on drought-stressed rice was higher than that in BPH feeding on non-stressed control plants. The size of myofibrils and the abundance of mitochondria in BPH flight muscles were significantly lower in BPH fed on drought-stressed rice compared to non-stressed plants. These results indicate that water management impacts the physiology of BPH, which may be useful in understanding the relationship between drought stress and this damaging herbivore.

## 1. Introduction

Crops are subject to attack by various abiotic and biotic factors during growth, which can lead to poor quality and reduced yields. Among the various abiotic stresses, drought is considered a critical threat to crop growth and sustainable agriculture worldwide [1]. Herbivores also constitute a source of biotic stress that can greatly impact crop growth and yield [2]. Continued efforts are required to identify more efficient and environmentally friendly methods for pest control. Therefore, elucidating changes in plant–herbivore interactions during drought stress is needed to reduce insect damage and determine the correct amount of pesticide needed for insect control [3]. 

Drought directly damages crops by causing intracellular water deficits, membrane damage and decreased enzymatic activity [4]. Drought also causes a redistribution of the ratio of primary metabolites to secondary metabolites, thereby changing the nutrition, palatability and resistance of host plants to herbivores [5]; these changes can subsequently affect the population growth of herbivores.

Similarly, the growth and development of herbivores is directly and indirectly influenced by heat and drought [6,7]. For example, the magnitude and frequency of heatwaves can directly affect the survival rate of insects [8] as they are especially vulnerable to water stress due to their small body size [9]. Water modulates many physiological functions in herbivores, including the absorption and transportation of nutrients and the maintenance of cellular turgor. Herbivores can absorb or excrete ions, amino acids and uric acid via fat bodies, digestive organs, and excretory organs and tissues, thus keeping osmotic pressure in the blood relatively stable [10]. Herbivores with piercing or sucking mouthparts ingest plant sap to maintain an appropriate water content and turgor pressure [11]; a decline in water content and a drop in hemolymph osmotic pressure can lead to physiological instability. 

In herbivores, various sugar molecules are involved in nutrient cycling, including glucose (a monosaccharide) and trehalose (a disaccharide) [12]. Herbivores also require amino acids to synthesize enzymes, transporters, structural proteins and receptor molecules [13]. Studies have shown that the activities of detoxification genes and enzymes are related to the presence of secondary compounds in plants. For example, glutathione S-transferase (GST), an important detoxification enzyme, can metabolize a large number of exogenous toxins, and helps insects adapt to stress [14,15]. During periods of drought stress, rice produces and accumulates reactive oxygen species, including O_2_^−2^, ·OH and H_2_O_2_, which destroy cellular structure [16]. These reactive molecules inhibit planthopper feeding and reproduction and increase insect mortality [17].

The brown planthopper (BPH; *Nilaparvata lugens* Stål) is the predominant pest of rice crops in China, and BPH damage is exacerbated by the monsoon-like climate in Southeast Asia [18,19,20]. The damaging effects of BPH are enhanced by its strong migratory and reproductive capacity. Migration is an insect behavior that may arise due to adverse environmental conditions [18]. Successful migration is dependent on suitable flight muscles; these are highly specialized fibrocytes, which are differentiated, mature muscle cells originating from myoblasts in vivo [21]. Insect flight muscles are rich in myofibrils and mitochondria, and are well-equipped to meet the energy demands of insect flight [21]. 

The effect of drought stress on BPH might be reflected in the development of the ovaries in female adults. Many genes impact ovary development in BPH, including *Exuperantia* (*Exu*), *Mago nashi* (*Mag*), *Vasa* (*Vas*) and *Vitellogenin* (*V**g*). The contribution of these genes to ovary development in insects has primarily been studied in *Drosophila melanogaster* [22]. For example, *Exu* ensures that the mRNA of the *bicoid* gene (*bcd*) is located in the proper position within the egg, and facilitates oocyte polarity [23,24]. *Mag* is a highly conserved gene that influences rear-pole polarity in *Drosophila*; thus, its expression contributes to the quality of reproduction [22]. *Vas*, a maternal effector gene, plays an important role in the formation of ventral segments and the development of germ cells. Vasa protein is the main component of generative substance, which is essential for the migration of generative cells [25]. The expression products of the *Vas* gene family are established and widely used molecular markers of germ cells. The *Vg* gene encodes the precursor of vitellogenin, which is mainly synthesized in the adipocytes and then released into the hemolymph of female insects. After selective absorption by oocytes, *Vg* is modified in the egg and finally deposited as vitellin (Vt), which provides the nutrients needed for embryonic development [26,27,28,29,30]. In female BPH adults, it has been shown that elevated levels of *Exu* expression reduce ovary development, and that *Exu* and *Mag* have similar functions, whereas *Vas* gene expression increased during ovary development [31]. 

Water scarcity has been shown to alter BPH feeding times and feeding sites on rice [32]; however, physiological changes in BPH after feeding on drought-stressed rice have not been studied. Research on drought stress in phytophagous insects has primarily focused on survival rates, growth, development, and fecundity [33]. We speculated that drought might also affect the physiological stability and flight ability of BPH. In this study, we measured changes in population numbers and alterations in the physiology and ultrastructure of BPH flight muscles after feeding on drought-stressed rice. Our results provide a basis for understanding physiological and population-wide responses in BPH during drought stress, and provide insights relating to pest control during periods of water scarcity. 

## 2. Materials and Methods

### 2.1. Experimental Biomaterials and Design

The wild-type rice cultivar Zhong Hua 11 (ZH11) was used in this study. ZH11 seeds were sown in pots and transplanted into plastic buckets; the latter were maintained in an environmentally controlled greenhouse at 26 ± 2 °C with a 16/8 h light–dark photoperiod and 70% RH for 10 d. 

BPH was collected from Yangzhou University (Yangzhou, China) and reared in the laboratory in an intelligent artificial climate chamber using the conditions described above. 

The experimental design contained two factorials with two elicitors: drought and non-drought conditions.

### 2.2. Drought Stress Treatments

Plants in the tillering stage (25 d) with similar growth and vigor were selected for subjection to moderate drought stress with a water potential of −30 to −25 kPa. Four barrels of plants (12 plants per barrel) were used as an experimental group. The drought- and non-drought-stressed plants were grown with a water layer (1–3 cm high) prior to the onset of experiments. During the experiments, drought-stressed plants were maintained under moderate drought stress, whereas the non-stressed plants were watered regularly. Water potential was recorded daily using a negative pressure gauge (Shenzhen Hengjinda Technology Co., Ltd., Shenzhen, China). When negative pressure exceeded the lower limit of water potential, plants were supplied with water to maintain conditions of moderate drought stress. 

### 2.3. BPH Population Numbers

Eight barrels of plants were divided into drought-stressed and non-stressed experimental groups. Rice plants from both drought-stressed and non-stressed treatment groups were randomly selected and covered with transparent plastic covers. Four fourth-instar nymphs (two males and two females) were released onto individual rice plants (12 plants per barrel). After nymphs developed into adults, the number of BPH per barrel was counted until the second generation of BPH adults began to emerge.

### 2.4. Impact of Drought Stress on the Expression of Ovarian Development Genes 

Total RNA was extracted from BPH females with the Promega SV Total RNA Isolation System (Vazyme Biotech Co., Ltd., Nanjing, China). Reverse transcription was performed with the PrimeScript^TM^ RT Reagent Kit with gDNA Eraser (Perfect Real Time, Vazyme Biotech Co., Ltd., Nanjing, China), and reverse transcription conditions were 37 °C for 15 min, 85 °C for 5 s, and 4 °C for 15 min. qRT-PCR was used to determine changes in mRNA levels. A three-step method was used with SYBR Premix Ex Taq^TM^ II (Tli RNaseH Plus) (TaKaRa Biotechnology Dalian Co., Ltd., Dalian, China), and each sample was replicated three times. The qRT-PCR amplification was conducted for 40 cycles as follows: 50 °C for 5 min, 95 °C for 5 s, 56 °C for 30 s, and 72 °C for 30 s. The qRT-PCR primers are listed in Table 1.

### 2.5. Anatomy of BPH Ovaries

Ovaries were collected from unmated BPH females raised on ZH11 rice. Ovaries were dissected and anatomical changes were described using the methods reported by Ge et al. [34]. 

### 2.6. Determination of Water Content in BPH

Second-generation BPH adults (*n* = 35) with similar body size were collected from the drought-stressed and non-stressed treatment groups, and fresh weight was determined. Insects were then transferred to an oven and incubated for 1 h at 105 °C. Insects were then removed, weighed, and allowed to cool for 30 min in closed Petri dishes to prevent changes in water content. Insects were then repeatedly dried and cooled until the weight difference of the same material before and after treatment was less than 0.002 g.

Water content was determined as the ratio of the difference between dry and fresh weights to total fresh weight, as follows:$$\mathrm{WC}=\frac{W_{1}-W_{2}}{W_{1}-W_{0}}\times 100\%$$

WC: water content of BPH (%); *W*_0_: weight of the empty Petri dish; *W*_1_: combined weight of the insect material and dish before drying; *W*_2_: combined weight of insect material and Petri dish after drying.

### 2.7. Determination of Osmotic Pressure in BPH Hemolymph

After the emergence of second-generation adults, BPH adults (*n* = 15) were collected from the drought-stressed and non-stressed treatment groups, and the osmotic pressure of the hemolymph was determined [35].

### 2.8. Glucose and Total Amino Acid Content in BPH

The determination methods were carried out according to the instructions of a total amino acid test kit (Nanjing Saihongrui Biotechnology Co., Ltd., Nanjing, China) and a glucose content kit (Beijing Solarbio Science & Technology Co., Ltd., Beijing, China), respectively.

### 2.9. Determination of Glutathione S-Transferase Activity in BPH

The determination method was carried out according to the Micro Glutathione S-transferase (GST) Assay Kit instructions (Beijing Solarbio Science & Technology Co., Ltd., Beijing, China).

### 2.10. Ultrastructure of Flight Muscles in BPH

Second-generation female adults were collected from each treatment group and stored in 2.5% glutaraldehyde solution at 4 °C. Seven days later, the insects were processed; heads and abdomen were removed with a dissecting needle, and the thorax was opened. The flight muscle in the thorax was removed using a microscope, transferred to 2.5% glutaraldehyde (0.1 mol/L phosphate buffer, pH 7.3), and stored at 4 °C. Four of the eight adult females were used to obtain a longitudinal section of the flight muscle, and the remaining four were used to obtain cross-sectional slices.

The flight muscle was soaked in 2.5% glutaraldehyde, washed three times with phosphate buffer, fixed in 1% osmium tetroxide for 2 h, rewashed three times with phosphate buffer, dehydrated with graded concentrations of ethanol (50, 70, 80, 90, 95 and 100%), and then washed with acetone. After soaking in a 1:1 mixture of acetone and resin for 1 h, the fight muscle samples was soaked in a 1:2 ratio of acetone and resin for 2 h, embedded in pure resin and then sliced with an EMUC6 frozen ultramicrotome (Leica Microsystems Ltd. Co., Wetzlar, Hesse-Darmstadt, Germany). The slices were stained with uranyl acetate and lead acetate, and then observed and photographed with a Tecnai 12 TEM (Philips-FEI Co. Ltd., Amsterdam, Holland) [36]. 

### 2.11. Analysis of Transmission Electron Micrographs

The diameter and cross-sectional area of flight muscle myofibrils, sarcomere length and percentage of mitochondria were measured by Image Pro Plus (Media Cybernetics Co. Ltd., Rockville, MD, USA).

The diameter of myofibril cross-sections was calculated as the mean value of the major and minor axes of individual myofibrils in the micrograph. The percentage of the cross-sectional area of the myofibrils was calculated by dividing the pixelation of each myofibril by the pixelation of the entire image. The cross-sectional area of the myofibril was obtained by multiplying the percentage of the cross-sectional area of the myofibrils by the area of the photo. The sarcomere length was calculated as the distance between two adjacent Z-lines in the cross-section of the myofibril. The percentage of mitochondria was calculated as the sum of mitochondrial pixels divided by the total pixels in the micrograph. Replicates containing at least eight myofibrils and ten micrographs of mitochondria were used to obtain each parameter [36].

### 2.12. Statistical Analysis 

Statistical significance between treatment groups was obtained by analysis of variance (ANOVA; Systat Inc., San Jose, CA, USA). Multiple comparisons were conducted using the Protected Least Significant Difference (PLSD) test. The data were denoted as means ± SE and analyzed using SPSS 16.0 software.

## 3. Results

### 3.1. Population Dynamics of BPH

When BPH was fed on drought-stressed rice for 15, 19 and 23 d, the population numbers were significantly lower than the population fed on non-stressed control plants (Figure 1). 45.74% (*F* = 30.09, df = 1, 7, *p* < 0.05), 36.77% (*F* = 19.25, df = 7, *p* < 0.05) and 24.61% (*F* = 9.70, df = 7, *p* < 0.05)

### 3.2. Effects of Drought Stress on the Fecundity of BPH

The expression of genes related to ovarian development was determined in BPH fed on rice seedlings exposed to drought stress (Figure 2). There was a significant increase in the expression levels of *NlVg* and *NlExu*, which were 12- and 1.4-fold higher than the control, respectively (Figure 2).

In BPH fed on drought-stressed and non-stressed rice, ovarioles contained two ripe-banana-shaped oocytes. There was no significant difference in ovary development and morphology in BPH female adults fed on drought-stressed and non-stressed rice when monitored at the same growth stage (Figure 3). 

### 3.3. Physiological Changes in BPH after Feeding on Drought-Stressed Rice

After moderate drought stress, the water content and osmotic pressure of BPH hemolymph were 18.80% and 25.10% lower, respectively, than in the non-stressed control group (*p* < 0.05), (Figure 4A,B). In response to drought stress, glucose content and GST activity increased by 149.26% and 10.06%, respectively, when compared to the control (Figure 4C,E). In contrast, the total amino acid content of BPH decreased by 66.03% under moderate drought stress (Figure 4D).

### 3.4. Ultrastructural Changes in BPH Flight Muscles after Feeding on Drought-Stressed Rice 

The length of flight muscle sarcomeres in drought-stressed and non-stressed BPH was significantly different (Figure 5 and Figure 6). The sarcomere length in BPH fed on drought-stressed rice was 26.54% shorter than the control (*F* = 246.70, df = 1,84, *p* < 0.05; Figure 5A,B and Figure 6A). Data from repeated general linear measurements indicated that myofibrils and mitochondria of BPH flight muscles were significantly affected by drought stress. Mitochondria in the flight muscles of BPH undergoing drought stress were 32.76% less abundant than in the non-stressed control group (*F* = 931.48, df = 1,19, *p*<0.05; Figure 5C,D and Figure 6B). Myofibril diameter was reduced by 29.38% (*F* = 92.50, df = 1,15, *p* < 0.05) and the cross-sectional area of myofibrils was reduced by 47.03% (*F* = 72.06, df = 1,19, *p* < 0.05) in drought-stressed BPH compared with the non-stressed controls (Figure 6C,D).

## 4. Discussion

Climate change has disrupted the global agricultural ecosystem and has caused changes in the ecological adaptability of insects, an increase in the number of generations, and the expansion of distribution areas [37]. Altered climate also negatively influences the nutritional composition, physiology and biochemistry of plants. Climate change may ultimately lead to alterations in plant–herbivore interactions [38]. Drought stress is one of the more injurious forms of climate change, and can negatively impact the growth and development of animals and plants [39]. Many studies have shown that prolonged drought stress can reduce herbivore reproduction capacity, and may lengthen the developmental period [40,41,42]. Furthermore, water-stressed plants cause a reduction in herbivore growth, survival, and abundance by inducing anti-herbivore defense systems in plants, or by reducing food quantity and nutritional quality [43,44,45]. To cope with reduced nutritional quality in host plants, phytophagous insects adapt by delaying their own growth and development [39]. In this study, the decrease in BPH population size after feeding on drought-stressed rice supports these observations.

Water content and the osmotic pressure of insect hemolymph are important characteristics of herbivorous insects undergoing drought stress. In order to survive, insects adjust their own water balance to adapt to environmental changes. Drought stress can also alter water metabolism in BPH by inducing changes in water content and enzyme activity in rice. Although continuous drought stress has a significant impact on insects [46], the effect of short-term drought stress on the growth and development of insects is less obvious than other abiotic stresses such as temperature and photoperiod. Our results showed that water content and the osmotic pressure of BPH hemolymph during continuous drought stress conditions were significantly lower than those in non-stressed control insects; furthermore, these negative effects on water balance and osmotic pressure can cause a decline in herbivore quality.

Carbohydrates are important for the growth and development of herbivorous insects [47], and sugars provide energy. Prior studies have shown that insects use sugars to produce energy and to regulate the osmotic potential of hemolymph, with the aim to achieve osmotic pressure equivalent to that of the sieve tubes in host plants [48]. There are also high concentrations of amino acids and derivatives in insect hemolymph, and these amino acids are used to synthesize enzymes, transporters, structural proteins and receptor molecules [13]. Several studies have found that the amino acid content in non-diapausing insects is higher than in diapausing insects; in the latter, amino acid levels gradually decrease with diapause length, which suggests that they are converted into other substances to maintain basic physiological functions [49]. In this study, the amino acid content of BPH was significantly lower in insects fed on drought-stressed rice compared to the non-stressed control group. It is possible that BPH might have difficulty obtaining adequate nutrition when feeding on drought-stressed rice, and amino acids might be diverted into other substances to sustain life. Signal transduction pathways involved in resistance to abiotic stress are activated in rice when infested by BPH. Flavonoid biosynthesis is one pathway that is frequently activated in plants subjected to abiotic stress such as drought [50]. Gong et al. [51] reported that flavone-treated BPH results in the production of salivary response proteins involved in the metabolism of energy, amino acids and carbohydrates, which leads to a decrease in amino acid content. These findings are consistent with our results.

When environmental parameters fluctuate, insects may induce the production of various endogenous or exogenous compounds to improve their adaptability. Detoxification enzymes play an important role in improving ecological adaptability and maintaining normal physiological and biochemical metabolism [16]. For example, insects may utilize GST to catalyze the binding of harmful substances with glutathione, which may help detoxify or eliminate harmful substances [52]. Insects may also reduce their food intake to avoid hazardous substances, or they may consume more sugars to produce the energy needed for detoxification. Our results showed that the GST content of BPH fed on drought-stressed rice was significantly higher than BPH fed on non-stressed control plants; this indicated that toxic substances related to insect resistance (e.g., reactive oxygen species) were present in drought-stressed rice, and that BPH-encoded GST played an important role in maintaining herbivore physiology.

In this research, the relative expression levels of *NlVg* and *NlExu* were higher in BPH fed on drought-stressed rice than in those feeding on non-stressed control plants. In female BPH adults, elevated levels of *Exu* expression reduces ovary development [31]. Our results showed that the relative expression of *NlExu* in BPH feeding on drought-stressed rice is higher, which means slower growth of ovaries. In the five developmental stages of female BPH ovaries, the relative expression of *Vg* increases and then decreases in the second and fifth stages, respectively [31]. In combination with Figure 3, the ovaries of the female BPH adults in the drought-stressed group were smaller than those of the control group. In summary, our results indicate that BPH fed on drought-stressed rice are more likely to exhibit altered development than to exhibit irreversible and destructive changes in herbivore ovaries.

Climate change, geography, rice cultivation and management measures, and the distribution of insects, are factors that impact the growth and migration of BPH. Climatic conditions directly affect the physiology and biochemistry of BPH through temperature and humidity, and also influence seasonal activities and long-distance migration due to changing environmental conditions in the field [53]. In this research, we compared the flight muscles of BPH fed on drought-stressed and non-stressed rice to examine the potential effect of drought on BPH migration. Generally, when myofibrils exhibit larger cross-sectional measurements and greater diameter, the flight muscles will be stronger and more resilient [36,37]. Our results showed that the length of flight muscle sarcomeres and mitochondrial content were decreased in BPH feeding on drought-stressed rice, which indicated that drought reduced flight ability. 

Our results provide compelling evidence that drought stress impacts BPH physiology. An interesting example is the shortened length of sacromeres in flight muscles when BPH was fed on drought-stressed rice. A controlled period of drought stress might be helpful to understand the relationship between drought stress and BPH infestation, and might be useful in reducing the distribution of BPH in the field; this further illustrates the value of this study in elucidating new avenues for BPH control and management. 

## Figures and Tables

**Figure 1 insects-13-00355-f001:**
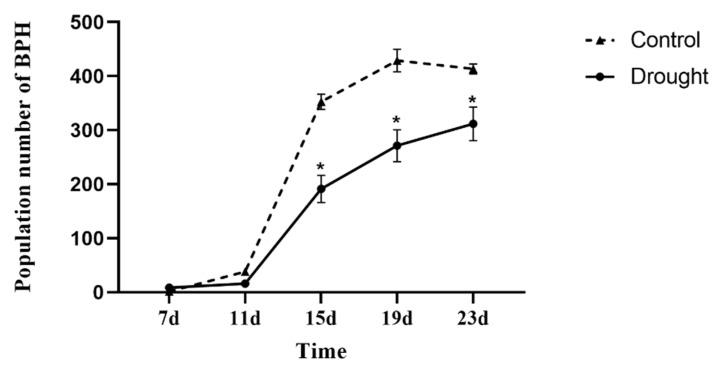
Population changes in BPH fed on drought-stressed and non-stressed (control) rice plants for 7, 11, 15, 19, and 23 days. Data are expressed as means ± SE. * represents significant differences between drought-stressed and control treatments (*PLSD* test, *p* < 0.05).

**Figure 2 insects-13-00355-f002:**
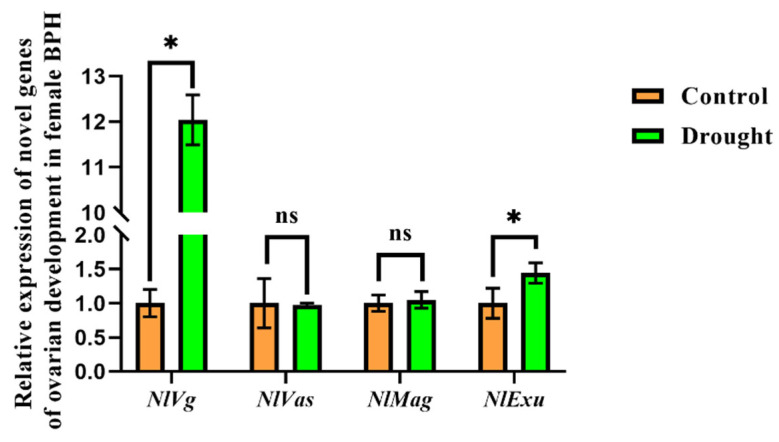
Relative expression of *NlVg*, *NlVas, NlMag* and *NlExu* in female BPH after feeding on drought-stressed and non-stressed rice. Data represent means ± SE. * represents significant differences in relative expression (*PLSD* test, *p* < 0.05).

**Figure 3 insects-13-00355-f003:**
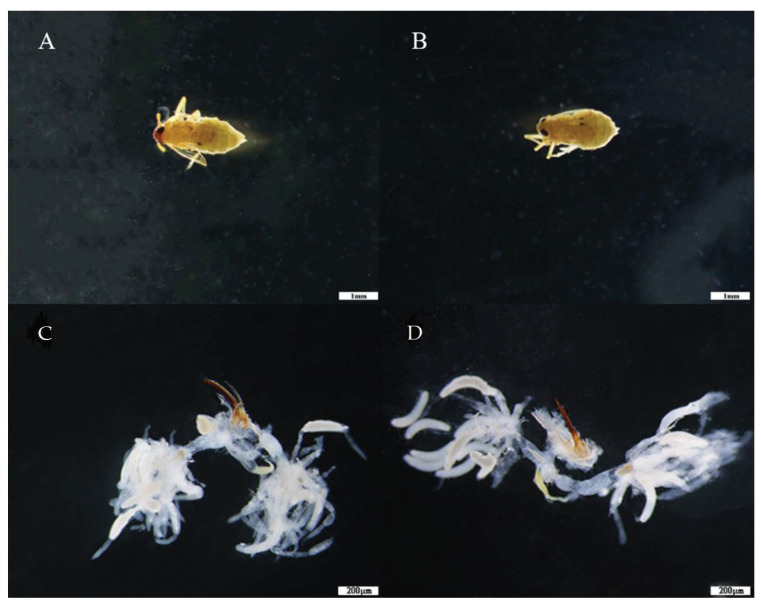
Morphology and ovary development in BPH fed on drought-stressed and non-stressed (control) rice. Panels (**A**,**a**) show morphology and ovary development in BPH feeding on non-stressed, control rice. Panels (**B**,**b**) show morphology and ovary development in BPH feeding on drought-stressed rice. (**A**,**B**) Body size was photographed with a Leica DMR connected to a Fuji FinePix S2 Pro digital camera (Wetzlar, Hesse-Darmstadt, Germany). Scale bar, 1 mm. (**a**,**b**) reproductive tracts from mated females; scale bar, 200 μm. Body size and ovaries from at least ten females were dissected and observed for each group.

**Figure 4 insects-13-00355-f004:**
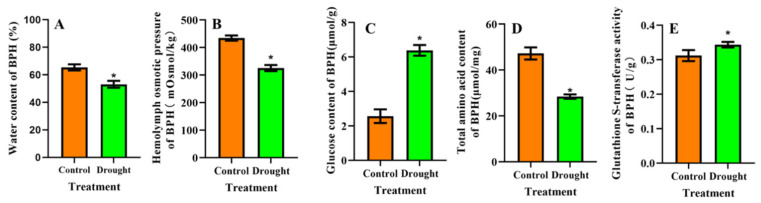
Physiological and biochemical changes in BPH fed on drought-stressed and non-stressed rice for 23 days. Changes in (**A**) water content, (**B**) osmotic pressure of hemolymph, (**C**) glucose content, (**D**) total amino acid content, and (**E**) GST activity. Data indicate means ± SE. * represents significant difference between drought-stressed and non-stressed BPH at *p* < 0.05 (*PLSD* test).

**Figure 5 insects-13-00355-f005:**
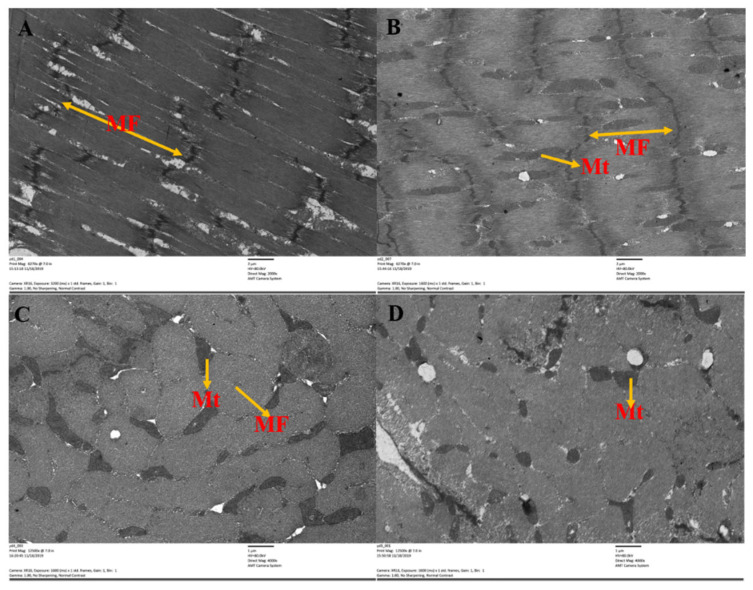
Longitudinal and cross-sectional views of flight muscles in BPH adult females fed on drought-stressed and non-stressed rice. Longitudinal section of flight muscles in (**A**) non-stressed female adults and (**B**) drought-stressed female adults (×6270). Cross-sectional view of flight muscles in (**C**) non-stressed female adults and (**D**) drought-stressed female adults (×12,500). Abbreviations: MF—myofibril; Mt—mitochondria.

**Figure 6 insects-13-00355-f006:**
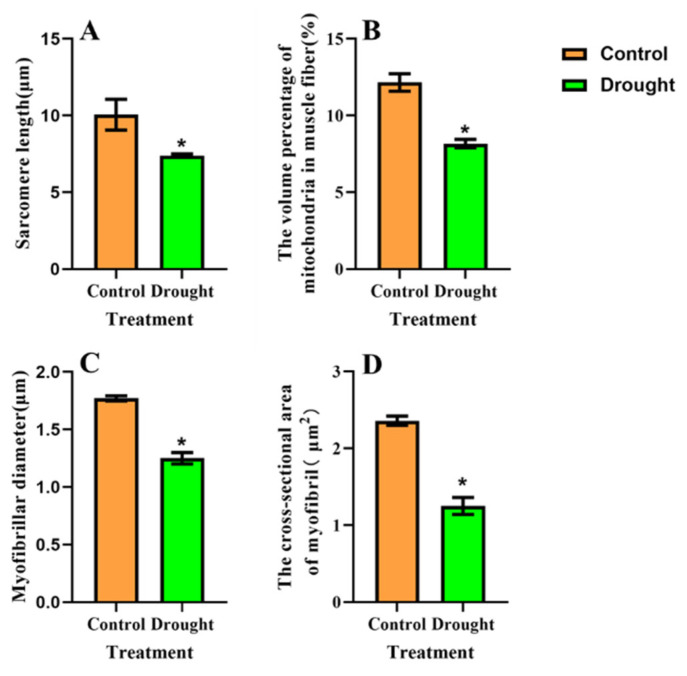
Changes in flight muscles of BPH adult females fed on drought-stressed and non-stressed rice: (**A**) sarcomere length; (**B**) percentage of mitochondrial volume in flight muscle fibers; (**C**) changes in the diameter of myofibrils; and (**D**) changes in the cross-sectional area of myofibrils. * represents significant differences at *p* < 0.05 (*PLSD* test).

**Table 1 insects-13-00355-t001:** qRT-PCR primers.

Primer Name	Sequences (5′–3′)	GenBankAccession No.
*NlActin*-qF	TGGACTTCGAGCAGGAAATGG	EU179846.1
*NlActin*-qR	ACGTCGCACTTCAGATCGAG	
*NlVg*-qF	GTGGCTCGTTCAAGGTTATGG	AB353856
*NlVg*-qR	GCAATCTCTGGGTGCTGTTG	
*NlExu*-qF	GGGCGCTCAGGGATAAGACT	DB823987
*NlExu*-qR	GGGCATCATGACAAAGCAGAA	
*NlMag*-qF	CCGGATGGCAAGTTGAGGTA	DB843362
*NlMag*-qR	TGTGGCCAAGGTGAGTCATCT	
*NlVas*-qF	CCGATGCGGTGGATGTACTC	CG3056
*NlVas*-qR	GGCGCTGCATCTCTTCAAGT

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
