# Peer review of "Physiological and Population Responses of Nilaparvata lugens after Feeding on Drought-Stressed Rice"

_insects, 2022, doi:10.3390/insects13040355_

Round 1

Reviewer 1 Report

This is a nice paper on the interactions between plant and insect physiology under drought stress. My only concern is that it is a bit exaggerated to think that stressing rice plants through drought might be used as a management tactic against BPH. The effect on the plant would surely be more severe. I would suggest that the authors might rethink the consequences of their study, which are surely important in modelling insect responses to drought at landscape and regional scales.

Reviewer 2 Report

The author has revised the manuscript completely according to the reviewer’s suggestions; therefore, I recommend the revised version of the manuscript can be accepted and published on Insects.

Reviewer 3 Report

The manuscript has been improved more better after the authors revised the manuscript carefully.   The questions havealso  been answered one by one.  Just as the authors described there was no significant change in the morphology and ovarian development in BPH between the drought stess treatment and control, though the images of rice  planthopper  and ovary in figure 3 are still not very clear.

Author Response

This manuscript is a resubmission of an earlier submission. The following is a list of the peer review reports and author responses from that submission.

Round 1

Reviewer 1 Report

I have read through the manuscript and feel that the authors have not adequately responded to the comments and suggestions from previous reviewers. However, a number of key requests were not addressed: removal of unnecessary figures, correct application of statistical tests, improvement of discussion. Furthermore, parts of the introduction continue to require extensive changes and the methods section remains poorly developed. This is a pity because the paper looks to have potential, but requires support from a statistician as well as a native English speaker, and must address shortcomings to the methodologies and logic of presentation. The paper as it currently stands is not suitable for publication in an international journal. I feel that the authors need to carefully revise their study, perhaps adding more experiments (specifically regarding oviposition and egg-laying), and prepare an improved manuscript, taking the necessary time to complete changes. 

Reviewer 2 Report

This is a revised version of a former manuscript, but the paper is still not acceptable. The methods used in the study are still poorly described and incomprehensible. E.g., drying animals for 1 h at 60°C will never lead to the dry weight of the animals. With the method used, authors will not get hemolymph (!) osmotic pressure. Description of measuring glucose and amino acid content, as well as GST activity, are totally inadequate.

The entire manuscript needs considerable editorial as well as language improvement. I have marked many of the inadequacies in the manuscript (e.g., authors measured changes in population quantity, not quality; why do they speak about "novel genes"? mmol/kg is a concentration, but not an osmontic pressure value; authors did not measure enzyme content, but enzyme activity; etc.)
